# Utility of a Card-Type Respiratory Rate Measuring Device for Spontaneously Breathing Patients

**DOI:** 10.3390/diagnostics15070864

**Published:** 2025-03-28

**Authors:** Yoshiaki Iwashita, Shun Takeda, Satoshi Kawashima, Shinya Sato, Satoru Nebuya

**Affiliations:** 1Department of Emergency and Critical Care Medicine, Faculty of Medicine, Shimane University, Ennyacho89-1, Izumo 693-8501, Shimane, Japan; 2Development, POSH WELLNESS LABORATORY Inc., 1-5-20 Moto-Akasaka, Minato-ku 107-0051, Tokyo, Japan; 3Department of Joint Research in Advanced Medicine for Electromagnetic Engineering, Shimane University, Ennyacho89-1, Izumo 693-8501, Shimane, Japan; 4Division of Rehabilitation Medicine, Shimane University Hospital, Ennyacho89-1, Izumo 693-8501, Shimane, Japan

**Keywords:** respiratory rate, heart rate, impedance measurement, contactless monitor, wearable sensors

## Abstract

**Background/Objectives:** Accurate measurement of respiratory rate (RR) is critical for early detection of patient deterioration. A newly developed contactless card-type device measures RR and heart rate (HR) by detecting chest impedance changes. Although previously validated in mechanically ventilated patients with 15% RR accuracy, its performance in spontaneously breathing patients remains uncertain. **Methods:** This prospective observational study was conducted at the Department of Emergency and Critical Care Medicine, Shimane University Hospital, in December 2022. Patients admitted to the emergency center without invasive mechanical ventilation were enrolled. The card-type device was subsequently placed on the left chest, while the RR and HR were simultaneously recorded with an electrocardiogram monitor. Data from both devices were collected and compared. **Results:** Six patients were enrolled. The RR measurements from the card-type device were within a 10% difference from those measured using the standard monitor in four out of six cases. In two cases, the card-type device recorded an RR lower than that of the standard monitor, which coincided with periods of patient speaking. For HR, the card-type device was within a 10% margin of the standard monitor in two cases, but it underestimated HR in the remaining four cases, particularly during high tidal volumes or increased thoracic thickness. **Conclusions:** The contactless card-type device accurately measured respiratory rates within a 10% margin compared to standard monitors in most non-intubated patients, except during activities such as speaking. Further studies with larger sample sizes are warranted to confirm these findings and improve the device’s performance.

## 1. Introduction

Some patients admitted to the general ward deteriorate after hospitalization due to conditions such as sepsis, respiratory failure, or cardiac events. To prevent such deterioration, a rapid response system (RRS) has been introduced in several hospitals in the US and Japan. The RRS is activated based on specific criteria, among which respiratory rate is considered a critical predictor of intensive care needs and poor patient outcomes [1,2,3]. However, in routine clinical practice, respiratory rate is commonly measured visually by healthcare providers, which is subjective and prone to inaccuracy. This often results in missed or incorrect measurements, affecting early detection of patient deterioration [4,5,6]. Therefore, efforts are ongoing in many hospitals to establish standardized and accurate respiratory rate measurements using automated or contactless methods [7,8].

POSH WELLNESS LABORATORY, Inc. (Tokyo, Japan) recently developed a contactless card-type respiratory and heart rate measuring device, which functions by detecting impedance changes in the patient’s left chest. Impedance-based monitoring offers a non-invasive and continuous method to assess respiratory effort, reducing observer bias. Previously, we attempted to validate the device in an emergency medical center and found that respiratory rate was measured within a 15% difference from mechanical ventilator monitoring [9]. However, four of five patients in that study were receiving invasive mechanical ventilation, limiting our ability to analyze respiratory rate in spontaneously breathing patients. Additionally, heart rate accuracy was suboptimal, likely due to signal interference or motion artifacts.

To address these limitations, we planned another pilot study to evaluate the performance of a modified version of the POSH card-type device in non-intubated patients. In the present study, we validated the use of a modified POSH card-type device to measure the respiratory rate and heart rate in non-intubated patients and compared it with a general respiratory rate measuring device. The aim of this study was to determine whether the card-type device accurately measures the respiratory rate and heart rate in non-intubated patients, as well as to reveal the situation in which the card-type device differs from common respiratory rate instruments.

## 2. Materials and Methods

This study was a prospective observational study conducted at the Department of Emergency and Critical Care Medicine in Shimane University Hospital in December 2022. This study was a pilot study, limited to a study period of one month, and was conducted without setting a target number of cases. The inclusion criteria were patients aged 20 years or older who visited the emergency room and were subsequently hospitalized in the Department of Emergency and Critical Care Medicine at Shimane University Hospital without undergoing invasive mechanical ventilation. The measurement process was performed by researchers from Shimane University, with technical assistance provided by the POSH WELLNESS LABORATORY Inc (Tokyo, Japan). The exclusion criteria included patients with an implanted pacemaker or any other similar electronic medical device that could interfere with impedance measurements. All participants in the study provided written informed consent prior to participation. This study was approved by the Shimane University Institutional Committee on Ethics (Study number 6280, approved on 17 May 2022).

### 2.1. Card-Type Device

The details of the card-type sensor are provided in a previous report [9]. In brief, the device developed by the POSH WELLNESS LABORATORY measures impedance changes between two points located on the dorsal side of the card. The key technological advantage of this device is its ability to perform non-contact impedance measurement of intra-thoracic cavity movements, distinguishing it from conventional ECG-based respiratory monitors that require electrode contact with the patient’s skin. The standard ECG-based monitors measure impedance changes that also include signals from chest wall muscle activity, which may introduce variability. The card sensor transmits collected data via Bluetooth to a computer where a dedicated application, pre-installed on the system, processes and analyzes the signals. The software automatically differentiates respiratory and heart rate signals based on amplitude fluctuations in impedance and displays real-time numerical values. Notably, the card-type sensor does not require direct skin contact, allowing it to be placed in a dedicated bag that is securely attached to the patient’s clothing (Figure 1). A significant improvement from previously reported devices is the enhanced heart rate identification sensitivity, achieved by modifying the algorithm responsible for detecting cardiac signals.

### 2.2. Study Protocol

Patients who met the inclusion criteria were admitted to the emergency center, and informed consent was obtained from the patient or their family. After consent was obtained, a card-type device was placed on the patient’s left chest. Patients were monitored using an IntelliVue electrocardiogram monitor (Philips Healthcare, Amsterdam, The Netherlands). The patient and IntelliVue monitor were recorded on video. The video and card sensor data were recorded simultaneously. Data from the card-type device were also recorded every second. Data from the monitor were extracted from the video every 10 s, and the patient’s body movements and speech activities were recorded. Data were analyzed for each patient. Since the monitor and the card-type sensor use different methods to measure respiratory rate and heart rate, we judged that a 10% error from the monitor was generally an accurate measurement, rather than seeking a complete agreement of the values. This value was set because the normal respiratory rate is 12 to 20 breaths, and a 10% difference would be 1 or 2 breaths, values that would not be considered a large difference from a clinical standpoint.

## 3. Results

Six patients were evaluated during the study period. The patient characteristics are listed in Table 1. Three of the patients were female, and three were male. The median age of the participants was 74 years, the median height was 156 cm, the median weight was 60.4 kg, and the median body mass index (BMI) was 24. Two patients were on non-invasive positive pressure ventilation (NPPV).

### 3.1. Respiratory Rate

Figure 2 presents a comparison of the respiratory rates measured by the card-type device and the usual monitor. In Cases 1, 3, 4, and 6, the respiratory rate in the card-type device was almost within the 10% limit of the usual monitor. In Case 2, the card-type device data were approximately 10/min lower than those of the usual monitor. The results from Case 5 were also lower for the card-type device than for the usual monitor. In such two cases, the respiratory rate on the usual monitor fluctuated significantly, whereas the data from the card-type device were more stable. From the video monitoring data, there was a long period of patients talking or attaching an ECG in Cases 2 and 5. The remaining patients did not move much during measurements.

### 3.2. Heart Rate

The participants’ heart rates are shown in Figure 3. The card-type device heart rates in Cases 5 and 6 were predominantly within the 10% limit of the usual monitor. The data for Cases 1–4 were lower than those for the usual monitor. The heart rate of the card-type Senso was highly fractalated in Case 1 and in Cases 2 and 5 during talking, whereas the usual monitor was stable. From the video monitor, Case 1 appeared to have a higher tidal volume of respiration. Case 2 involved a female with a high BMI and thickened thorax.

## 4. Discussion

In this study, we conducted a prospective observational validation of a contactless card-type respiratory rate and heart rate measuring device in non-intubated patients. We further found that the device could measure respiratory rates within a 10% error margin compared with standard monitoring equipment in most cases. Our results also demonstrated that, even in non-intubated patients, the card-type device showed high accuracy at measuring respiratory rates. The respiratory rate data differed significantly between the card-type device and the usual monitor during patient speech or body movement. Regarding heart rate, although the sensitivity of the card sensor improved from the previous model, inaccuracies were noted in patients with a high tidal volume or increased thoracic thickness.

It is generally known that unexpected adverse events occur in 10–15% of patients admitted to medical facilities [10,11,12]. In recent years, an increasing number of hospitals have introduced RRS to establish patient safety in hospitals, in which a team intervenes early in the case of relatively minor deterioration of a patient’s condition. Recent studies on RRS have focused on respiratory rate as one of the indicators for early detection of sudden hospital emergencies [13,14,15]. The respiratory rate is an important vital sign when predicting patient deterioration, but it remains poorly measured, even in hospital settings [4,5,6]. Furthermore, even when respiratory rates are measured, inter-observer differences exist among medical professionals, such as physicians and nurses [16,17,18]. Currently, many healthcare providers are educated to recommend measuring respiratory rates. These educational activities are said to improve the frequency of respiratory rate measurements [19]. However, the standard method of measuring respiration in modern medical care is to visually check the movement of the thorax for 20 s and multiply it by three to determine the number of breaths per minute. This method requires the patient and the person measuring to stop talking or moving and observe the thorax movement for a while, which is time-consuming and may cause the patient to become conscious of breathing, which may result in a change from the usual respiratory rate. Therefore, accurate wearable respiratory rate monitoring is needed, particularly for patients who are not critically ill and do not require a wired device, such as a medical ECG monitor. Furthermore, during the COVID-19 pandemic, wearable vital monitoring systems are in demand, and many devices have been developed and assessed [20,21,22,23]. However, most of the wearable monitors are wearable but need contact with skin. Patients entering a general ward are able to take care of themselves, and being wired to a monitor is stressful. In recent years, non-contact respiratory rate measurement devices have also been developed, but these use a camera to monitor thoracic movements [24,25]. It might be suitable for detecting the sleep apnea syndrome during sleep at home [26,27]; however, it is not suitable for continuous monitoring of patients admitted to general wards. The primary strength of this device is that it is a contactless wearable device that can be placed on a name card, or in a breast pocket, to allow for remote vital monitoring. As such, it can be used for a wide range of applications, from monitoring critically ill patients in hospitals to patients in general wards and emergency rooms, as well as patients in home care. Our device has already demonstrated accuracy in measuring respiratory rate in ICU settings [9]. This study adds a possible application for patients in the general ward.

In the present study, we noted that the respiratory rate data from the card-type sensor differed significantly from the usual monitor when patients were speaking. It is difficult to count the respiratory rate during a conversation, as speaking induces chest wall movements. There have been several studies on respiratory patterns during conversation and exercise in healthy volunteers or lung disease patients [28,29,30,31,32]. In those studies, the respiratory pattern was assessed by a flow sensor with a face mask; no studies assessed the respiratory rate by an ECG-based monitor. Since a usual ECG-based monitor counts respiratory rate by counting the impedance change in the chest wall, it also counts the chest wall movement without respiration, such as conversation and exercise. The key significance of the POSH-produced device lies in its ability to filter out noise from normal conversational breathing because only the gas exchange in the lung would be counted in our device. This makes the card-type sensor potentially more accurate during patient interactions than traditional ECG-based respiratory rate monitoring. In fact, from the video monitor images, the actual respiratory rate appeared to be more accurate on the card-type sensor than on the regular ECG monitor. The current study did not include measurement of respiratory frequency as a survey item, nor was it possible to accurately re-measure respiratory frequency with the video monitor. Further research is needed to determine the gold standard of respiratory rate during conversation or exercise.

For heart rate monitoring, although the device exhibited limitations in detecting variations during high tidal volumes, it generally performed well. Due to the characteristics of the card-type sensor used in this study, it is not suitable for non-contact measurement of impedance changes from the body surface in people with thick rib cages and large ventilation volume. For heart rate, there are a number of wristband-type measurement devices on the market, and they are highly accurate [33]. In addition, since wearing a wrist-type sensor is not a burden for people who wear a wristwatch in their daily lives, the validity of measuring heart rate with a wristband-type sensor may be better established than with a card-type sensor on the front of the chest wall. The strength of the card-type sensor marks a significant advancement towards improving patient monitoring in non-critical settings, offering a potential tool for the early detection of patient deterioration through counting the respiratory rate without the need for intrusive equipment.

This is the first study to validate this contactless impedance-based card-type respiratory rate monitor in non-intubated patients. The strength of this study lies in its practical application in a clinical setting and the demonstration of the high accuracy and ease of use of our device. However, this study has some limitations; one is a small sample size and short study duration. Large-scale studies including multiple institutions are needed. Second, the difficulty in validating which monitor is more accurate when patients are talking or moving. We also should plan to compare the ECG-based respiratory rate and card-type sensor respiratory rate by measuring the flow sensor with a facemask as a gold standard. Additionally, the performance of the device during high tidal volume breathing and in patients with a thickened thorax requires further investigation. For heart rate monitoring, compared to wristband-type devices, future studies should address these limitations by exploring larger and more diverse patient populations to confirm these findings.

## 5. Conclusions

In conclusion, the present study validated the efficacy of a novel contactless card-type respiratory rate and heart rate measuring device in non-intubated patients. The device accurately measured respiratory rates within a 10% margin of error compared with standard monitors, except during patient speech. Heart rate measurement is less reliable in cases of high tidal volume or increased thoracic thickness. This device is a promising tool for the early detection of patient deterioration in non-critical settings, thereby contributing to improved patient care and monitoring.

## Figures and Tables

**Figure 1 diagnostics-15-00864-f001:**
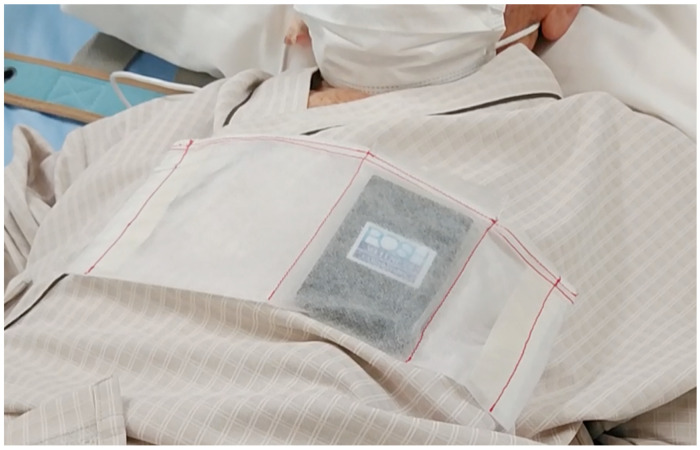
Card-type sensor being attached. A card-type sensor is being placed in a special bag on the patient’s left anterior chest. The device is not necessary to be placed in the bag. We did this because the hospital cloth did not have a breast pocket.

**Figure 2 diagnostics-15-00864-f002:**
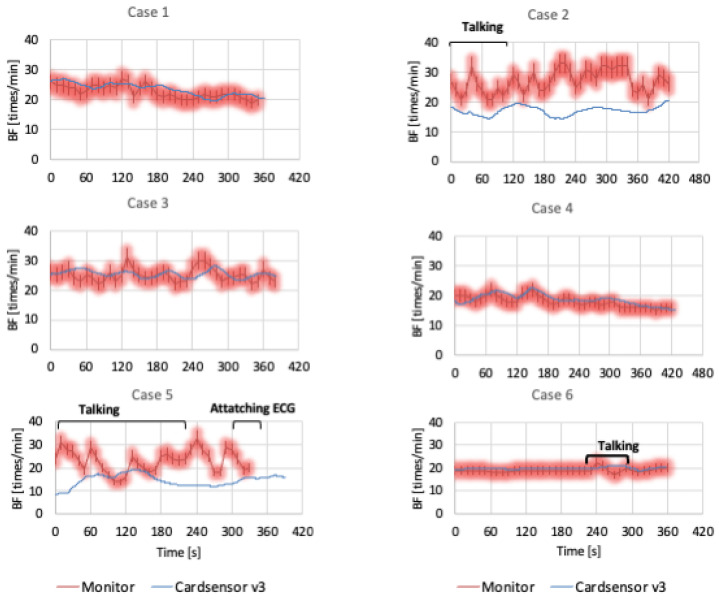
Comparison of the respiratory rate in the card-type device and usual monitor. The blue line represents the respiratory rate on the card-type device, while the red bar presents the ±10% deviation from the usual monitor value. From the video monitor, we identified that the patient was talking and attached it to a 12-lead ECG. ECG: electrocardiogram, BF: breath frequency.

**Figure 3 diagnostics-15-00864-f003:**
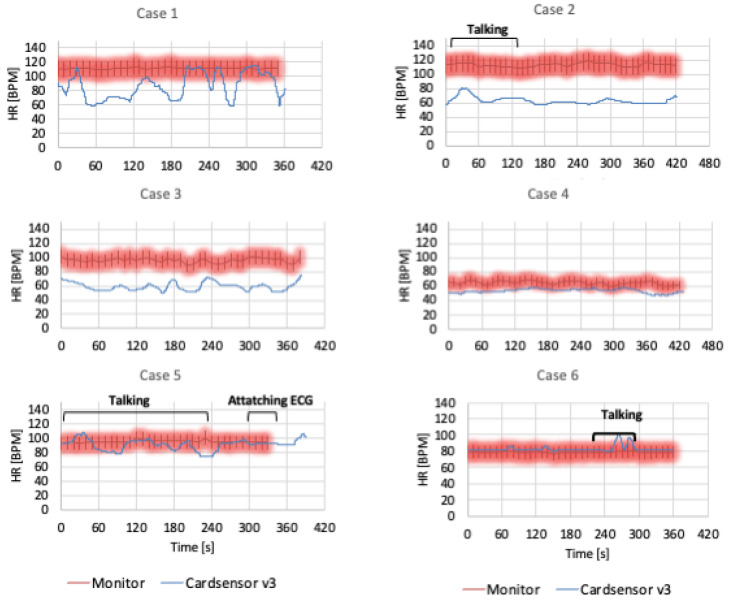
Comparison of the heart rate in the card-type device and usual monitor. BPM: beats per minute.

**Table 1 diagnostics-15-00864-t001:** Patient characteristics.

	Age	Sex	Height (cm)	Weight (kg)	BMI	Diagnosis
1	61	F	155	40.8	17	Pneumonia, on NPPV
2	38	F	166	70	25	Left tubal pyosalpinx
3	97	F	145	42	20	Right femur fracture
4	83	M	157	66.4	27	Renal failure, pulmonary edema, on NPPV
5	78	M	154	54.3	23	Esophageal stricture
6	70	M	167	70	25	Lung cancer

F: female, M: male, BMI: body mass index, NPPV: non-invasive positive pressure ventilation.

## Data Availability

All other data are available from the corresponding author on reasonable request.

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
