# Peer review of "Utility of a Card-Type Respiratory Rate Measuring Device for Spontaneously Breathing Patients"

_diagnostics, 2025, doi:10.3390/diagnostics15070864_

Round 1
Reviewer 1 Report
Comments and Suggestions for Authors
Dear Editor and Authors,
Thank you for your submission and dear Editor thank you for asking me to review this manuscript titled "Utility of a Card-Type Respiratory Rate Measuring Device for Spontaneously Breathing Patients".
In this work the authors describe a novel contactless, card-type device to meassure
respiratory rate (RR) and heart rate (HR) by detecting chest impedance changes. They have previously tested this device in ICU patients and now they are doing so in spontaneously breathing ones. This device is in its very early development testing with a number of issues been raised. In the ICU patients due to mechanical ventilation and artifact the device was not able to provide accurate meassurements. Now, the authors present the results of their pilot study in spontaneously breathing/awake patients.
However, even this time the results are still sub-optimal with only a 15% accuracy achieved!! The reported 10% within the monitor limit is arbitrary and quite large in range to be honest!
I have the following comments to make:
- I am unsure why the device has to be placed in a cloth bag to work and how this is practical!! I guess this is for the development department to sort out but it seems a bit clumsy!
- You mention data are transmited via bluetooth to a computer. Do you plan to minimize this into a smaller poket portable device?
- Since 2 patients were on positive pressure ventilation then this would taint your data!! Why did you not exclude these patients and include 2 other ones which where spontaneously and unassisted breathing??
- If the patients where speaking then why couldn't the meassurements be taken when they were quite or ask them to be quiet!!
- There seem to be quite a lot of limitations of this device!! It doesn't work if the patient is speaking, if he is in NPPV, is she has a high BMI!! Where do you see this device been applicable at and will this improve?
- Did the authors caclulate expected and actual sensitivity of the device?
- Why 6 patients and not 8 or 10? Was there a time limitation to recruitment? Why not perform a better patient selection instead of including patients with conditions/habitus that would impede the data!!
- How can the authors claim in their discussion section that the device was accurate in an ICU setting when in their introduction they mentioned that invasive ventilation and artifact was a problem?? I mean this device has no use in an ICU setting since monitoring can be performed the traditional way but it does have a use in awake/active patients!!
- How will you be able to monitor an active patient who moves about and changes the location of the device in terms of space - area chest (i.e. when he stands up or walks vs sitting down or laying down!!).
Minor Issues:
- Correct affiliation and address is needed! Where is Shimane University and POSH WELLNESS LABORATORY Inc located. Please fix!
In conclusion, this pilot study is too limitted and problematic even for a pilot study!! There are a number of methodological issues and quite a bit of improvement that can occur!! Thank you.
Comments on the Quality of English LanguageNeeds some minor language editing but nothing tragic.
Reviewer 2 Report
Comments and Suggestions for Authors
I have carefully read this manuscript that I was commissioned to review. It is a pilot study that evaluates the efficacy and feasibility of a contactless card-type device for monitoring RR and HR, comparing it with traditional monitoring systems. The work appears well written:
- The introduction provides a broad and clear background of the type of device being studied and clarifies the purposes of this pilot study;
- The methods appear clear and rigorous; likewise the results, expressed with a simplicity that makes it difficult for the reader to be distracted;
- The discussion is well written and offers a careful analysis of the existing literature on the topic.
Therefore, I congratulate the authors. However, I have some concerns:
1) why, given the physiopathological non-selectivity and the simplicity of application and reading of the monitoring system, does the study have such a low sample size?
2) With respect to the rationale of the study, which sees RR as a predictive marker of clinical deterioration in any condition, there is a contradiction in applying a system that does not require contact and therefore can be subject to easy movements in a conscious subject who maintains motor and language skills
3) Do you think that the conclusions should be less clear-cut. It cannot be said that a system works if in just over 50% of the sample size it detects a variability of less than 10% compared to conventional systems.
In any case, it is a good work and I believe that with these small refinements it can be safely published
Round 2
Reviewer 1 Report
Comments and Suggestions for Authors
Dear Editor and Authors,
I have reviewed and re-evaluated this revised manuscript again and I considered the changes and answers the authors have given. The work still has a lot of limitations some of which can't be corrected at this stage (for example sample size, ect).
However, since this is a pilot, experimental study requirements are less vigorous and thus I am now content to accept it for publication given the changes made. Kind regards.
Comments on the Quality of English LanguageNeeds some language polishing up!